# Transformers in DLOGTIME-**Uniform** $\mathsf{TC}^0$

**David Chiang**                                                          *dchiang@nd.edu*
*University of Notre Dame*

**Reviewed on OpenReview:** *https://openreview.net/forum?id=ZA7D4nQuQF*

## Abstract

Previous work has shown that the languages recognized by average-hard attention transformers (AHATs) and softmax-attention transformers (SMATs) are within the circuit complexity class $\mathsf{TC}^0$. However, these results assume limited-precision arithmetic: using floating-point numbers with $O(\log n)$ bits (where $n$ is the length of the input string), Strobl showed that AHATs can be approximated in L-uniform $\mathsf{TC}^0$, and Merrill & Sabharwal showed that SMATs can be approximated in DLOGTIME-uniform $\mathsf{TC}^0$. Here, we improve these results, showing that AHATs with no approximation, SMATs with $O(\mathsf{poly}(n))$ bits of floating-point precision, and SMATs with at most $2^{-O(\mathsf{poly}(n))}$ absolute error are all in DLOGTIME-uniform $\mathsf{TC}^0$.

## 1 Introduction

Previous work (summarized in Table 1) has shown that the languages recognized by average-hard attention transformers (AHATs) and softmax-attention transformers (SMATs) are within the circuit complexity class $\mathsf{TC}^0$. This places some interesting computational problems beyond the power of these transformers. In particular, if $\mathsf{TC}^0 \neq \mathsf{NC}^1$ (as is often assumed, Williams 2022), then these transformers cannot solve any $\mathsf{NC}^1$-complete problems. For example, consider Boolean formulas with constants $0$ and $1$ and no variables, like $(0 \wedge \neg 1) \vee (\neg 0 \wedge 1)$. Checking the *syntax* of such formulas is equivalent to the Dyck language, which is recognizable by both AHATs (Yao et al., 2021) and SMATs (Yang & Chiang, 2024). But computing the *semantics* of such formulas, that is, deciding whether a formula is true, is $\mathsf{NC}^1$-complete (Buss, 1987) and therefore not solvable by these transformers (unless $\mathsf{TC}^0 = \mathsf{NC}^1$).

However, these non-solvability results assume limited-precision arithmetic. The best results that we are aware of use floating-point numbers with $O(\log n)$ bits (where $n$ is the length of the input string): Strobl (2023) showed that AHATs can be approximated in L-uniform $\mathsf{TC}^0$, and Merrill & Sabharwal (2023b) showed that SMATs can be approximated in DLOGTIME-uniform $\mathsf{TC}^0$. These results leave open the possibility that AHATs and SMATs, as defined on paper using real numbers, might not be subject to the same limitations. Here, we improve these results, showing that:

- AHATs (without any approximation) are in DLOGTIME-uniform $\mathsf{TC}^0$.

- SMATs with $O(\mathsf{poly}(n))$ bits of floating-point precision are in DLOGTIME-uniform $\mathsf{TC}^0$.

Furthermore, because there are many different ways to approximate a transformer using limited precision, and different ways appear to lead to different results, we propose an alternative assumption, which is that the final output is approximated up to a certain (absolute) error. Thus, we show:

- SMATs with at most $2^{-O(\mathsf{poly}(n))}$ absolute error are in DLOGTIME-uniform $\mathsf{TC}^0$.

This can also be rephrased as a statement, not about the expressivity of approximations of SMATs, but about the expressivity of exact SMATs themselves:

- Any language that is recognized by a SMAT with margin $2^{-O(\mathsf{poly}(n))}$ is in DLOGTIME-uniform $\mathsf{TC}^0$.

Table 1: Summary of results on transformer encoders in previous work and in this paper. Our results show that even when (average-hard attention or softmax-attention) transformers are computed to very high precision, they remain limited to DLOGTIME-uniform $\mathsf{TC}^0$.

| | attention | approximation | class |
|---|---|---|---|
| Merrill et al. (2022) | average | $O(\log n)$ precision | non-uniform $\mathsf{TC}^0$ |
| Liu et al. (2023) | softmax | $O(\log n)$ precision | non-uniform $\mathsf{TC}^0$ |
| Strobl (2023) | average | $O(\log n)$ precision | L-uniform $\mathsf{TC}^0$ |
| Merrill & Sabharwal (2023a) | softmax | $O(\log n)$ precision | L-uniform $\mathsf{TC}^0$ |
| Merrill & Sabharwal (2023b) | softmax | $O(\log n)$ precision | DLOGTIME-uniform $\mathsf{TC}^0$ |
| This paper, Theorem 7 | average | none | DLOGTIME-uniform $\mathsf{TC}^0$ |
| This paper, Theorem 13 | softmax | $O(\mathsf{poly}(n))$ precision | DLOGTIME-uniform $\mathsf{TC}^0$ |
| This paper, Theorem 14 | softmax | $2^{-O(\mathsf{poly}(n))}$ error | DLOGTIME-uniform $\mathsf{TC}^0$ |

## 2 Background

We write $[n]$ for the set $\{1, 2, \ldots, n\}$. We write $\lfloor x \rfloor$ for the floor of $x$ (greatest integer less than or equal to $x$), and $\lceil x \rceil$ for the ceiling of $x$ (least integer greater than or equal to $x$). We write $O(\mathsf{poly}(n))$ for the family of functions $\bigcup_{k \geq 0} O(n^k)$.

### 2.1 Transformers

We assume familiarity with transformers (Vaswani et al., 2017) and describe a few concepts briefly. For more detailed definitions, please see the survey by Strobl et al. (2024), whose notation and terminology we follow.

In standard attention, attention weights are computed from attention scores using a softmax:

$$\alpha_{i,j} = [\text{softmax } s_{i,*}]_j = \frac{\exp s_{i,j}}{\sum_{j'} \exp s_{i,j'}}.$$

We call a transformer with standard attention a *softmax-attention* transformer (SMAT). An *average-hard attention* transformer (AHAT, Pérez et al. 2019; Merrill et al. 2022) is one where the softmax is replaced by:

$$\text{ahardmax } s_{i,*} = \lim_{\tau \to 0} \text{softmax } s_{i,*}/\tau.$$

In other words, each position $i$ attends to those positions $j$ that maximize the score $s_{i,j}$. If there is more than one such position, attention is divided equally among them.

*Layer normalization* (Ba et al., 2016) scales and shifts the components of a vector to have mean and standard deviation equal to parameters $\gamma$ and $\beta$:

$$\text{LayerNorm}(\mathbf{x}) = \frac{\mathbf{x} - \text{E}[\mathbf{x}]}{\sqrt{\text{Var}[\mathbf{x}] + c}} \odot \gamma + \beta \tag{1}$$

where $\odot$ is componentwise multiplication and $c \geq 0$ is a constant. When layer normalization is used, we require that $c > 0$ (as is standard in practice).

We assume that a transformer has a single scalar output, computed from the last position. For simplicity, we assume that the output is used for binary classification, as follows:

**Definition 1.** A transformer $T \colon \Sigma^* \to \mathbb{R}$ recognizes a language $L$ if, for every string $w \in \Sigma^*$, if $w \in L$ then $T(w) > 0$, and if $w \notin L$ then $T(w) < 0$.

## 2.2 Complexity classes

A $\mathsf{TC}^0$ circuit is one with made from the usual AND, OR, NOT gates, as well as MAJORITY gates, which are true if a strict majority of their inputs are true. A $\mathsf{TC}^0$ circuit family is a set of circuits indexed by lengths $n > 0$, such that the circuit for length $n$ has polynomial size, bounded depth, and unbounded fan-in. DLOGTIME-uniform $\mathsf{TC}^0$ is the set of $\mathsf{TC}^0$ circuit families for which queries about the circuit for length $n$ can be decided in deterministic $O(\log n)$ time. Throughout this paper, whenever we say $\mathsf{TC}^0$, we mean DLOGTIME-uniform $\mathsf{TC}^0$.

The class $\mathsf{TC}^0$ is also the class of languages definable in first-order logic with majority quantifiers ($\mathrm{M}x.\phi(x)$ iff $\phi(x)$ is true for a majority of positions $x$) and the BIT predicate ($\mathrm{BIT}(x, y)$ iff the $y$-th bit of $x$ is 1) (Barrington et al., 1990). Depending on the context, it may be easier to think about $\mathsf{TC}^0$ in terms of circuits or in terms of logical formulas. Our descriptions of functions in $\mathsf{TC}^0$ abstract away from details of either circuits or formulas, making use of functions already known to be in $\mathsf{TC}^0$ together with the fact that functions in $\mathsf{TC}^0$ are closed under serial and parallel composition (Jeřábek, 2012).

**Theorem 2.** *The following operations on $O(\mathsf{poly}(n))$ bit integers are in $\mathsf{TC}^0$:*

*(a) Addition of two numbers*

*(b) Comparison of two numbers*

*(c) Maximum of $n$ numbers*

*(d) Truncated base-2 logarithm $\lfloor \log_2 x \rfloor$*

*(e) Iterated addition of $n$ numbers*

*(f) Multiplication of two numbers*

*(g) Iterated multiplication of $n$ numbers*

*(h) Truncated division of two numbers.*

*Proof.* Addition (a) is shown by Immerman (1999, Prop. 1.9) for $n$ bits and is easy to extend to $O(\mathsf{poly}(n))$ bits. Comparison (b), maximum (c), and truncated base-2 logarithm (d) are also easy. These cases do not require majority gates.

Iterated addition (e) is shown, for example, by Barrington & Maciel (2000, Lecture 7, Section 2), and multiplication (f) is closely related.

Iterated multiplication (g) was proven to be in $\mathsf{TC}^0$ by Hesse et al. (2002, Theorem 5.1) and can be used for truncated division (h). □

## 2.3 Approximation error

We will define various numeric representations and associated concepts as they are needed, but will make use of the following definitions throughout.

**Definition 3.** For functions $\hat{f} \colon X \to \mathbb{R}$ and $f \colon X \to \mathbb{R}$, we say that $\hat{f}$ approximates $f$ with absolute error at most $\epsilon$ if for all $x \in X$, we have $|\hat{f}(x) - f(x)| \le \epsilon$, and $\hat{f}$ approximates $f$ with relative error at most $\epsilon$ if for all $x \in X$, we have $\left| \frac{\hat{f}(x) - f(x)}{f(x)} \right| \le \epsilon$.

# 3 Arbitrary-precision AHATs

In this section, we prove that AHATs without layer normalization, even with arbitrary precision, are in $\mathsf{TC}^0$. We do this by representing rational numbers as pairs of integers. This turns out to only need a polynomial number of bits, so it can be computed in $\mathsf{TC}^0$.

**Definition 4.** A *p-bit rational number* is a pair $\langle a, b \rangle$, where $a$ is an integer in $[-2^p, 2^p)$ and $b$ is an integer in $[1, 2^p)$. The *value* of $\langle a, b \rangle$ is $a/b$.

(According to this definition, a *p*-bit rational number actually requires $(2p+1)$ bits: 1 for the sign, $p$ for the numerator, and $p$ for the denominator.)

**Lemma 5.** *The following operations on $O(\mathsf{poly}(n))$-bit rational numbers are in $\mathsf{TC}^0$:*

- *(a) Addition, multiplication, division, and comparison of two numbers*
- *(b) Iterated multiplication of n numbers*
- *(c) Iterated addition and maximum of n numbers.*

*Proof.* The operations (a,b) can be expressed in terms of operations on $O(\mathsf{poly}(n))$-bit integers, which are in $\mathsf{TC}^0$ (Theorem 2):

$$\langle a_1, b_1 \rangle + \langle a_2, b_2 \rangle = \langle a_1 b_2 + b_1 a_2, b_1 b_2 \rangle \tag{2}$$

$$\langle a_1, b_1 \rangle \times \langle a_2, b_2 \rangle = \langle a_1 a_2, b_1 b_2 \rangle \tag{3}$$

$$\langle a_1, b_1 \rangle \div \langle a_2, b_2 \rangle = \langle a_1 b_2, b_1 a_2 \rangle \tag{4}$$

$$\langle a_1, b_1 \rangle \leq \langle a_2, b_2 \rangle \Leftrightarrow a_1 b_2 \leq b_1 a_2 \tag{5}$$

$$\prod_{i \in [n]} \langle a_i, b_i \rangle = \left\langle \prod_{i \in [n]} a_i, \prod_{i \in [n]} b_i \right\rangle. \tag{6}$$

To find the sum or maximum of $n$ rational numbers (c), we precompute the product of the denominators:

$$B = \prod_{j \in [n]} b_j$$

$$\sum_{i \in [n]} \langle a_i, b_i \rangle = \left\langle \sum_{i \in [n]} a_i B / b_i, B \right\rangle \tag{7}$$

$$\max_{i \in [n]} \langle a_i, b_i \rangle = \left\langle \max_{i \in [n]} a_i B / b_i, B \right\rangle. \tag{8}$$

$\square$

**Lemma 6.** *Let $T$ be an $\mathsf{AHAT}$ with rational weights, p-bit position embeddings, and no layer normalization. Let $L$ be the depth of $T$. Then the computation of $T$ needs $O(pn^L)$ bits for each intermediate and final value.*

*Proof.* First, note that if $\langle a_1, b_1 \rangle$ uses $O(n^k)$ bits and $\langle a_2, b_2 \rangle$ uses $O(n^k)$ bits, then their sum, product, and quotient (Eqs. (2) to (4)) also use $O(n^k)$ bits. But if $\langle a_i, b_i \rangle$ for $i \in [n]$ use $O(n^k)$ bits each, then their sum (Eq. (7)) uses $O(n^{k+1})$ bits.

We prove the lemma by induction on $L$. If $L = 0$, we just look up the embeddings, which need $O(p)$ bits per value. If $L > 0$, assume that layer $(L - 1)$ required $O(pn^{L-1})$ bits per value. In the self-attention, the queries, keys, values, and scores need $O(pn^{L-1})$ bits. The sum of the maximum-scoring values, which there could be up to $n$ of, needs $O(pn^L)$ bits, as does the average. Finally, the activations of the FFNN also need $O(pn^L)$ bits. $\square$

**Theorem 7.** *Let $T$ be an $\mathsf{AHAT}$ with rational weights, $O(\mathsf{poly}(n))$-bit position embeddings, and no layer normalization. Then the language recognized by $T$ is in $\mathsf{TC}^0$.*

*Proof.* $\mathsf{AHAT}$s use only the operations in Lemma 5 on rational numbers with $O(\mathsf{poly}(n))$ bits (Lemma 6). Since these operations are all computable in $\mathsf{TC}^0$ and can be composed in $\mathsf{TC}^0$, the language recognized by $T$ is in $\mathsf{TC}^0$. $\square$

**Remark 8.** We now have a more or less complete characterization of which regular languages can be recognized by AHATs. Barrington et al. (1992) showed that every regular language $L$ is either in $\mathsf{ACC}^0$ or $\mathsf{NC}^1$-complete.

- If $L$ is in $\mathsf{ACC}^0$, then it can be defined in linear temporal logic with modular counting (Baziramwabo et al., 1999), and therefore it can be recognized by an AHAT with suitable position encodings (Barceló et al., 2024).

- If $L$ is $\mathsf{NC}^1$-complete, then by Theorem 7 it cannot be recognized by an AHAT unless $\mathsf{TC}^0 = \mathsf{NC}^1$.

## 4 Polynomial-precision SMATs

Next, we turn to SMATs, extending Merrill & Sabharwal's proof from $O(\log n)$ bits to $O(\mathsf{poly}(n))$ bits.

**Definition 9.** A *p-bit floating-point number* is a pair $\langle m, e \rangle$ where $m$ (called the *significand*) and $e$ (called the *exponent*) are integers, $|m| \in \{0\} \cup [2^{p-1}, 2^p)$, and $e \in [-2^p, 2^p)$. The *value* of $\langle m, e \rangle$ is $m \cdot 2^e$. We write $\mathrm{round}_p(x)$, where $x$ is either a real number or a floating-point number, for the $p$-bit floating-point number nearest to $x$. If there are two such numbers, we call $x$ a *breakpoint* and define $\mathrm{round}_p(x)$ to be the one with an even significand.

(According to this definition, a $p$-bit floating-point number actually requires $(2p + 2)$ bits: $(p + 1)$ for the significand and its sign, and $(p + 1)$ for the exponent and its sign.)

To compute a SMAT with $p$-bit floating-point numbers means to approximate the operations in the SMAT with operations on floating-point numbers. In typical floating-point implementations, addition, multiplication, division, and square root are rounded to the nearest floating-point number, but exp is only approximated with a relative error of about $2^{-p}$. We also assume that summation of $n$ numbers is performed exactly and then rounded (following Liu et al. 2023; Chiang et al. 2023; Merrill & Sabharwal 2023a; but *pace* Li et al. (2024), who argue that rounding should be performed after each addition).

**Lemma 10.** *The following operations on floating-point numbers with $p \in O(\mathsf{poly}(n))$ bits are computable in $\mathsf{TC}^0$, with exact rounding to the nearest $p$-bit floating-point number:*

*(a) Addition, multiplication, division, and comparison of two numbers*

*(b) Iterated multiplication of $n$ numbers.*

*Proof.* These operations on $O(\mathsf{poly}(n))$-bit integers are in $\mathsf{TC}^0$ (Theorem 2). We just have to show that they are also definable on floating-point numbers. This is not a new result, but we try to fill in some details here that are missing elsewhere.

First, $\mathrm{round}_p(\langle m, e \rangle)$ can be computed in $\mathsf{TC}^0$ as follows: Count the number of significand bits $q = \lfloor \log_2 |m| \rfloor + 1$ (Theorem 2d), shift $m$ right by $(q - p)$ bits, and increment $e$ by $(q - p)$. Round $m$ to the nearest integer, and if $|m| = 2^p$, shift $m$ and increment $e$ once more. For the operations (a), we have

$$\langle m_1, e_1 \rangle + \langle m_2, e_2 \rangle = \begin{cases} \mathrm{round}_p(\langle m_1 + m_2 \mathbin{/\!/} 2^{e_1 - e_2}, e_1 \rangle) & \text{if } e_1 \geq e_2 \\ \mathrm{round}_p(\langle m_1 \mathbin{/\!/} 2^{e_2 - e_1} + m_2, e_2 \rangle) & \text{if } e_1 \leq e_2 \end{cases}$$

$$\langle m_1, e_1 \rangle \times \langle m_2, e_2 \rangle = \mathrm{round}_p(\langle m_1 m_2, e_1 + e_2 \rangle)$$

$$\langle m_1, e_1 \rangle \div \langle m_2, e_2 \rangle = \mathrm{round}_p(\langle m_1 \cdot 2^{p-1} \mathbin{/\!/} m_2, e_1 - e_2 - p + 1 \rangle)$$

$$\langle m_1, e_1 \rangle \leq \langle m_2, e_2 \rangle \Leftrightarrow \begin{cases} m_1 \leq m_2 \mathbin{/\!/} 2^{e_1 - e_2} & \text{if } e_1 \geq e_2 \\ m_1 \mathbin{/\!/} 2^{e_2 - e_1} \leq m_2 & \text{if } e_1 \leq e_2. \end{cases}$$

The operation $\mathbin{/\!/}$ is defined as

$$a \mathbin{/\!/} b = \begin{cases} a/b & \text{if } a/b \text{ is a multiple of } 1/4 \\ a/b + 1/8 & \text{otherwise.} \end{cases}$$

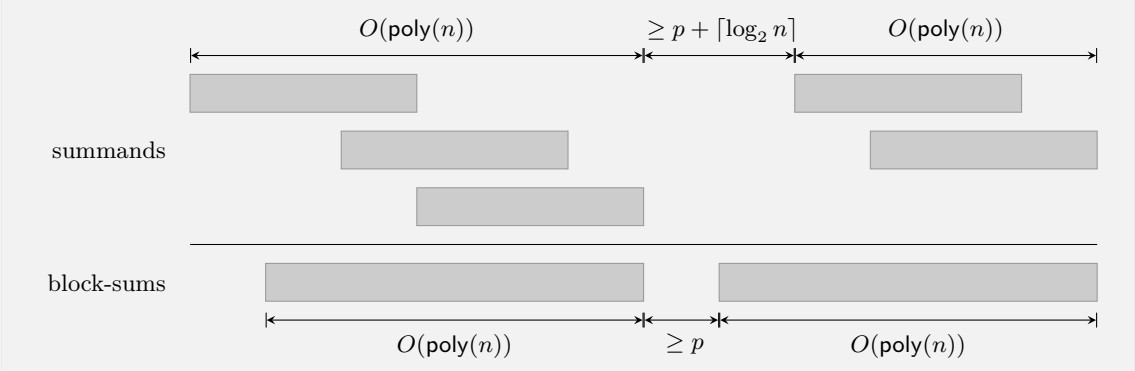

Figure 1: Overview of algorithm for iterated addition of $p$-bit floating-point numbers. The summands are grouped into blocks that each span $O(\mathsf{poly}(n))$ bits. They are separated by at least $p + \lceil \log_2 n \rceil$ bits, so that the block-sums are separated by at least $p$ bits.

The result has three fractional bits (called the *guard*, *round* and *sticky* bits), which ensure that the result is correctly rounded to the nearest floating point number (Goldberg, 2017). Note that this can be computed efficiently even if $b$ is a large power of 2.

For iterated multiplication (b), we have

$$\prod_{i \in [n]} \langle m_i, e_i \rangle = \mathrm{round}_p \left( \left\langle \prod_{i \in [n]} m_i, \sum_{i \in [n]} e_i \right\rangle \right).$$

$\square$

**Lemma 11.** *Iterated addition of $n$ floating-point numbers, each with $p \in O(\mathsf{poly}(n))$ bits, is in $\mathsf{TC}^0$.*

*Proof.* We are given $p$-bit floating-point numbers $\langle m_1, e_1 \rangle, \dots, \langle m_n, e_n \rangle$. Without loss of generality, assume $m_i \neq 0$. We need to compute the sum

$$s = \mathrm{round}_p \left( \sum_{i \in [n]} \langle m_i, e_i \rangle \right).$$

*Step 1.* Define the relation $i \sim j$ just in case $|e_i - e_j| < 2p + \lceil \log_2 n \rceil$. The transitive closure of $\sim$ partitions the (indices of the) summands into blocks $B_1, \dots, B_k \subseteq [n]$ (that is, if $i \sim j$, then $i$ and $j$ are in the same block). The intuition (Fig. 1) is that, in the binary representation, the numbers within each block are close enough together that we can sum them by brute force, while numbers in different blocks are far enough apart that we can ignore all but the two leftmost blocks.

The partitioning into blocks can be computed in $\mathsf{TC}^0$ as follows. Call $i \in [n]$ *block-minimal* iff there is no $j \in [n]$ such that $\langle m_j, e_j \rangle < \langle m_i, e_i \rangle$ and $i \sim j$. Then $i$ and $j$ belong to the same block if and only if there is no block-minimal $k \in [n]$ such that $e_i < e_k \leq e_j$ or $e_j < e_k \leq e_i$.

*Step 2.* For each block $B$, we compute the sum of all the numbers in $B$. Let $e$ be the minimal exponent in $B$ (that is, $e = \min_i \{e_i \mid i \in B\}$). Since all the exponents in $B$ are bigger than $e$ by at most $n(2p + \lceil \log_2 n \rceil) \in O(\mathsf{poly}(n))$, we can perform this sum exactly (Merrill & Sabharwal, 2023a):

$$\sum_{i \in B} \langle m_i, e_i \rangle = \left\langle \sum_{i \in B} m_i \cdot 2^{e_i - e}, e \right\rangle.$$

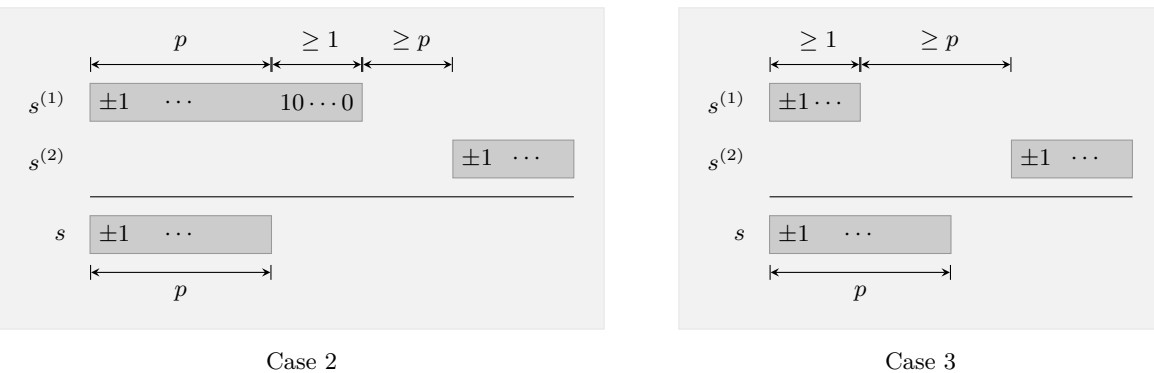

Case 2            Case 3

Figure 2: In Case 2, $s^{(1)}$ is a breakpoint, so the sum $s$ depends on the sign (and only the sign) of $s^{(2)}$. In Case 3, even if $m^{(1)}$ has only a single bit, the remaining block-sums do not affect the whole sum.

We've left the block-sums unnormalized; that is, their significands could have more or less than $p$ bits.

*Step 3.* Let $s^{(i)} = \langle m^{(i)}, e^{(i)} \rangle$ be the sum of the block with the $i$-th largest absolute sum. Then the first block-sum $s^{(1)}$ dominates the whole sum; any number not in the first block has absolute value less than $\langle 2^p, e^{(1)} - 2p - \lceil \log_2 n \rceil \rangle$. So we can bound the rest of the sum as:

$$r = \sum_{i=2}^{k} s^{(i)} < n \cdot 2^p \cdot 2^{e^{(1)} - 2p - \lceil \log_2 n \rceil} \leq 2^{e^{(1)} - p}. \tag{9}$$

In other words, in the binary representation, there is a gap of at least $p$ zero bits between the first block-sum and the remaining block-sums.

It's not necessary to sort all the block-sums; it's enough to find the maximal block-sum $s^{(1)}$ and the second block-sum $s^{(1)}$. Then we consider three cases (see Fig. 2).

Case 1: If $m^{(1)} = 0$, then the whole sum is zero, and we are done.

Case 2: If $s^{(1)}$ is a breakpoint, then we need to look at the remainder $r$ to see which way to round. Since $r < 2^{e^{(1)}}$ (Eq. (9)), it's enough to look at the sign of $r$, which is the sign of $m^{(2)}$.

Case 3: Otherwise, $s^{(1)}$ is sufficiently far (on the number line) from a breakpoint that the addition of $r$ cannot change the result. Due to cancellation, $m^{(1)}$ could have fewer than $p$ bits, down to just one bit. So the distance to the nearest breakpoint could be as small as $2^{e^{(1)} - p}$. But $r < 2^{e^{(1)} - p}$ by Eq. (9). □

**Lemma 12.** *Given a floating-point number $x$ with $O(\mathsf{poly}(n))$ bits, the following functions can be computed in $\mathsf{TC}^0$:*

    *(a) $\sqrt{x}$, rounded to the nearest floating-point number*

    *(b) $\exp x$, with a relative error of at most $2^{-p}$.*

*Proof.* The basic idea is to use a truncated Taylor series (Merrill, p.c.; Hesse et al., 2002, Cor. 6.5). This is not a new result, but we try to fill in some details here that are missing elsewhere. Let $p \in O(\mathsf{poly}(n))$.

For $\sqrt{x}$: Find $r \in [\frac{1}{4}, 1]$ and an even integer $k$ such that $x = r \cdot 2^k$, as follows. If $x = \langle m, e \rangle$ and $e + p$ is even, let $r = m \cdot 2^{-p} \in [\frac{1}{2}, 1)$ and $k = e + p$; if $e + p$ is odd, let $r = m \cdot 2^{-p-1} \in [\frac{1}{4}, \frac{1}{2})$ and $k = e + p + 1$. Then compute $\sqrt{r}$ using the Taylor series about 1:

$$\sqrt{r} = \sum_{i=0}^{N-1} \binom{\frac{1}{2}}{i} (r-1)^i + O(|r-1|^N).$$

Since the error term is in $O(|r-1|^N)$ and $r \geq \frac{1}{4}$, there is some $a$ such that the error is at most $a|r-1|^N \leq a\left(\frac{3}{4}\right)^N$. To make this less than $2^{-p-1}$, we set $N = \frac{(p+1)\log 2 + \log a}{-\log \frac{3}{4}} \in O(p)$. Then we decide which way to round by squaring the breakpoint nearest to the approximation of $\sqrt{r}$ and comparing it with $r$. Finally, $\sqrt{x} = \sqrt{r} \cdot 2^{k/2}$.

For $\exp x$: Let $k = \lfloor x/\log 2 \rfloor$ and $r = x - k\log 2$, where $\log 2$ is computed using the series:

$$\log 2 = \sum_{i=1}^{N-1} \frac{1}{i \cdot 2^i} + O(2^{-N}).$$

Compute $\exp r$ using the Taylor series about 0:

$$\exp r = \sum_{i=0}^{N-1} \frac{r^i}{i!} + O(r^N).$$

Since the error term is in $O(r^N)$ and $r \in [0, \log 2)$, there is some $a$ such that the relative error is at most $\frac{ar^N}{\exp r} \leq a(\log 2)^N$. So to get a relative error of $2^{-p}$, we set $N = \frac{p\log 2 + \log a}{-\log\log 2} \in O(p)$. Finally, $\exp x = (\exp r) \cdot 2^k$. $\qquad\square$

**Theorem 13.** *Any language that is recognizable by an $O(\mathsf{poly}(n))$-bit precision* SMAT *is in* $\mathsf{TC}^0$.

*Proof.* SMATs use only the operations in Lemmas 10 to 12. Since these operations are all computable in $\mathsf{TC}^0$ and can be composed in $\mathsf{TC}^0$, the language recognized by an $O(\mathsf{poly}(n))$-bit precision SMAT is in $\mathsf{TC}^0$. $\qquad\square$

## 5 Approximating SMATs with $2^{-O(\mathsf{poly}(n))}$ error

Defining "transformers with $p$-bit precision" and characterizing the class of languages they recognize is complicated, because there are many different ways to perform rounding, which can lead to differences in expressive power (Li et al., 2024). In this section, we propose an alternative approach, which is to limit the error of the final result of a transformer approximation and abstract away from details (like precision and rounding) of how that level of error is achieved. We show that approximating a SMAT with absolute error at most $2^{-O(\mathsf{poly}(n))}$ can be done in $\mathsf{TC}^0$.

This has two advantages. First, it has a simple and unambiguous definition. Second, it will allow us to say something about the expressivity of a large subclass of exact SMATs, namely, those that accept or reject strings with margin $2^{-O(\mathsf{poly}(n))}$.

**Theorem 14.** *For any* SMAT *$T: \Sigma^* \to \mathbb{R}$ and for any $\epsilon(n) \in 2^{-O(\mathsf{poly}(n))}$, there is a function $\hat{T}: \Sigma^* \to \mathbb{R}$ in $\mathsf{TC}^0$ such that for all $w \in \Sigma^*$ with $n = |w|$, $|\hat{T}(w) - T(w)| \leq \epsilon(n)$.*

*Proof.* We construct $\hat{T}$ out of the following operations, where $C > 0$ and $c > 0$ do not depend on $n$:

(a) Addition of two numbers

(b) Multiplication $xy$ where $|x|, |y| \leq C$

(c) Comparison of two numbers

(d) Inverse square root $\frac{1}{\sqrt{x}}$ where $|x| \geq c$

(e) Iterated addition of $n$ numbers

(f) Softmax of $n$ numbers.

The upper bound $C$ on all activations was shown by Hahn (2020), and in operation (d), the lower bound $c$ exists because we defined layer normalization to add a constant to the variance (Eq. (1)).

To simplify the error analysis, all of the above operations are performed on $O(\mathsf{poly}(n))$-bit rational numbers. In $\mathsf{TC}^0$, all of these operations can be computed exactly (Lemma 5), except $\sqrt{x}$ and $\exp x$, which can be approximated with relative error $\epsilon$ for any $\epsilon \in 2^{-O(\mathsf{poly}(n))}$, by Lemma 12. In that Lemma, the case for square root asks for $r \in [\frac{1}{4}, 1]$ and an even integer $k$ such that $x = r \cdot 2^k$. We do this as follows. If $a \geq b$, compute $\lfloor \frac{a}{b} \rfloor$ using truncated division (Theorem 2h), then count the number of bits (Theorem 2d) to get $k = \lfloor \log_2 \lfloor \frac{a}{b} \rfloor \rfloor + 1$. Similarly, if $a < b$, compute $k = -\lfloor \log_2 \lfloor \frac{b}{a} \rfloor \rfloor + 1$. Finally, if $k$ is odd, increment it by 1.

Fix $\epsilon_{\mathrm{final}} > 0$. We show by induction that, for each operation $i$ in the computation of $\hat{T}$, there is a $\delta_i \in \Theta(\epsilon/\mathsf{poly}(n))$ such that if we compute operation $i$ with error $\delta_i$, then the final answer has error $\epsilon_{\mathrm{final}}$. In particular, it is possible to compute $\hat{T}$ using $O(\mathsf{poly}(n))$-bit rationals and achieve a final error of at most $2^{-O(\mathsf{poly}(n))}$.

For each operation, we will show that for any $\epsilon > 0$, there is a $\delta \in \Omega(\epsilon/n)$ such that if the inputs to the operation are approximated with error $\delta$, then the output is approximated with error $\epsilon$.

If a function $f \colon \mathbb{R}^d \to \mathbb{R}$ is $\rho$-Lipschitz continuous, then for any $\epsilon > 0$, if $\|\mathbf{h}\| \leq \epsilon/\rho$, then $|f(\mathbf{x} + \mathbf{h}) - f(\mathbf{x})| \leq \rho\|\mathbf{h}\| \leq \epsilon$. Operations (a–d) are $\rho$-Lipschitz continuous with $\rho$ not depending on $n$, while iterated addition of $n$ numbers (e) is $n$-Lipschitz continuous, and softmax of $n$ numbers (f) is $\rho$-Lipschitz continuous with $\rho$ not depending on $n$.

We show the cases of inverse square root and softmax, as these also have error due to the Taylor approximations.

For inverse square root $y = \frac{1}{\sqrt{x}}$ for $x \geq c$: For any $\epsilon > 0$, let $\delta = \min\left(\frac{c}{2}, \frac{c\sqrt{c}}{(2c+1)\sqrt{2}}\epsilon\right)$. Suppose that $x$ has been approximated as $\hat{x} = x + h$ where $|h| \leq \delta$. Because $\sqrt{x}$ for $x \geq c - h \geq \frac{c}{2}$ is $\frac{1}{\sqrt{c}}$-Lipschitz continuous, we have $|\sqrt{\hat{x}} - \sqrt{x}| \leq \frac{\delta}{\sqrt{c}}$. Furthermore, we approximate $\sqrt{\hat{x}}$ with relative error $\eta$ where $|\eta| \leq \delta$. So we approximate $y$ as $\hat{y} = \frac{1}{\sqrt{\hat{x}}(1+\eta)}$, and the error is

$$
\begin{aligned}
|\hat{y} - y| &= \left| \frac{1}{\sqrt{\hat{x}}(1+\eta)} - \frac{1}{\sqrt{x}} \right| \\
&\leq \left| \frac{1}{\sqrt{\hat{x}}(1+\eta)} - \frac{1}{\sqrt{\hat{x}}} \right| + \left| \frac{1}{\sqrt{\hat{x}}} - \frac{1}{\sqrt{x}} \right| && \text{triangle inequality} \\
&= \left| \frac{\eta}{\sqrt{\hat{x}}(1+\eta)} \right| + \left| \frac{\sqrt{x} - \sqrt{\hat{x}}}{\sqrt{\hat{x}x}} \right| \\
&\leq \frac{\delta}{\sqrt{\frac{c}{2} \cdot \frac{1}{2}}} + \frac{\frac{\delta}{\sqrt{c}}}{\sqrt{\frac{c}{2} \cdot c}} && \eta \leq \delta, \hat{x} \geq c, \hat{x} \geq \frac{c}{2} \\
&= \frac{(2c+1)\sqrt{2}}{c\sqrt{c}}\delta \\
&\leq \epsilon.
\end{aligned}
$$

For softmax of $n$ numbers: For any $\epsilon > 0$, let $\delta = \min\left(\frac{1}{2}, \frac{\epsilon}{16}\right)$. Suppose that for all $i \in [n]$, $x_i$ has been approximated as $x_i + h_i$ where $|h_i| \leq \delta$, and let $\eta_i$ where $|\eta_i| \leq \delta$ be the relative error of approximating $\exp(x_i + h_i)$. Then the softmax and its approximation are

$$
y_i = \frac{\exp x_i}{\sum_j \exp x_j}
$$

$$
\hat{y}_i = \frac{(\exp(x_i + h_i))(1 + \eta_i)}{\sum_j (\exp(x_j + h_j))(1 + \eta_j)}
$$

and $\hat{y}$ overestimates $y$ by at most

$$
\begin{aligned}
\hat{y}_i - y_i &\leq \frac{(\exp(x_i + \delta))(1 + \delta)}{\sum_j (\exp(x_j - \delta))(1 - \delta)} - y_i \\
&= \left( \frac{(\exp 2\delta)(1 + \delta)}{1 - \delta} - 1 \right) y_i \\
&\leq \frac{(\exp 2\delta)(1 + \delta)}{1 - \delta} - 1 && y_i \leq 1 \\
&\leq \frac{(1 + 4\delta)(1 + \delta)}{1 - \delta} - 1 && 2\delta \in [0, 1] \Rightarrow \exp 2\delta \leq 1 + 4\delta \\
&\leq \frac{8\delta}{1 - \delta} && \delta \leq \tfrac{1}{2} \\
&\leq 16\delta && \delta \leq \tfrac{1}{2} \\
&\leq \epsilon && \delta \leq \tfrac{\epsilon}{16}.
\end{aligned}
$$

Similarly, we can show that $\hat{y}$ underestimates $y$ by at most

$$
y_i - \hat{y}_i \leq 8\delta \leq \epsilon. \qquad \square
$$

The above is a statement about the expressivity of SMAT approximations, but as mentioned at the beginning of this section, it also makes it possible to say something about the expressivity of a large subclass of exact SMATs.

**Definition 15.** A transformer $T \colon \Sigma^* \to \mathbb{R}$ recognizes a language $L$ with margin $\epsilon(n)$ if, for every string $w \in \Sigma^*$ with $n = |w|$, if $w \in L$ then $T(w) > \epsilon(n)$, and if $w \notin L$ then $T(w) < -\epsilon(n)$.

**Corollary 16.** *Any language that is recognizable by a SMAT with margin $2^{-O(\mathsf{poly}(n))}$ is in $\mathsf{TC}^0$.*

*Proof.* Let $L$ be a language recognized by SMAT $T$ with margin $\epsilon \in 2^{-O(\mathsf{poly}(n))}$. By Theorem 14, there is a function $\hat{T}$ in uniform $\mathsf{TC}^0$ such that for all $w$, we have $-\epsilon \leq \hat{T}(w) - T(w) \leq \epsilon$. If $w \in L$, then $T(w) > \epsilon$, so $\hat{T}(w) \geq T(w) - \epsilon > 0$. Similarly, if $w \notin L$, then $T(w) < -\epsilon$, so $\hat{T}(w) \leq T(w) + \epsilon < 0$. Therefore, $\hat{T}$ also recognizes $L$. $\qquad \square$

## 6 Limitations and Conclusions

The levels of precision considered here go far beyond what is practical to compute with. Nevertheless, these results are valuable because they further strengthen the case that transformers cannot compute any function outside of $\mathsf{TC}^0$.

Moreover, Section 5 offers an alternative approach to limited-precision transformers that may be useful in more realistic settings. In particular, an analogous argument shows that it takes $O(\log n)$ bits of precision to achieve an error of $1/O(\mathsf{poly}(n))$, which may make SMATs with margin $1/O(\mathsf{poly}(n))$ an interesting target for future research.

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
