# OpenReview forum: "Transformers in Uniform TC$^0$"
_TMLR — Accepted by TMLR_

### Review · Reviewer_gNug · 2024-11-19

**Summary Of Contributions:**

This paper explores the circuit complexity of Average-Hard Attention Transformers (AHATs) and Softmax-Attention Transformers (SMATs), demonstrating that both architectures belong to the complexity class $TC^0$ under the assumption of high-precision arithmetic. The authors first prove that AHATs, when equipped with position embeddings of $O(\text{poly}(n))$-bit precision and without layer normalization, can be computed within DLOGTIME-uniform $TC^0$. This result is derived through a detailed analysis of the operations involved in AHATs, leveraging existing results on the circuit complexity of these operators. Similarly, the paper shows that all operations in SMATs, using $O(\text{poly}(n))$-bit precision, also fall within DLOGTIME-uniform $TC^0$. Finally, the authors introduce a new approximation technique based on the absolute error, proving that SMATs with an error bound of at most $2^{-O(\text{poly}(n))}$ are computable within DLOGTIME-uniform $TC^0$.

**Audience:**

Yes

**Claims And Evidence:**

Yes

**Requested Changes:**

- I am not very familiar with the complexity circuits of transformers. However, I have noticed that some prior works also omit layer normalization for simplicity in their proofs. Could the authors clarify whether your results still hold when layer normalization is included?

- I am also unsure whether the absolute error bound of $2^{-O(\text{poly}(n))}$ is too loose in relation to the existing theorems. Could the authors provide additional insights or potential strategies to further tighten this bound?

**Strengths And Weaknesses:**

**Strength**
- The paper is generally well-written, and clearly structured.
- The main theorems are well-motivated and intuitive, focusing on the new high-precision scenario and improving previous results.
- Introducing absolute error as a new approximate method adds a fresh perspective to transformer complexity analysis.

**Weakness**
- As acknowledged by the authors, the assumptions of $O(\text{poly}(n))$-bit precision and $2^{-O(\text{poly}(n))}$ absolute error are highly unrealistic in practical settings. This limits the applicability of the theoretical results to provide meaningful insights for real-world transformer implementations.

---

> ### Author Response · Authors · 2024-11-19
> **Layer normalization**
>
> Thank you for your review. Theorem 7 does not allow layer normalization (because the square root would introduce irrational numbers), while Theorems 13 and 14 do allow layer normalization. Theorem 14 requires that layer normalization be defined as in section 2.1 with $c>0$ (as is standard in practice).

---

> ### Author Response · Authors · 2024-11-19
> **Is $2^{-O(poly(n))}$ too loose?**
>
> To answer this question, maybe it's helpful to observe that in previous work on limitations of transformers, there are two opposite tendencies: one is to keep the definition of transformers as close as possible to the original definition (which used real numbers), and the other is to make the definition as close as possible to actual implementations (which use fixed-precision floating-point numbers). This paper falls squarely in the first category. Presumably, more accuracy is more difficult to attain, and this paper shows that even with an unrealistically high level of accuracy, transformers are still computationally limited. From this point of view, I interpret "tight" to be "more accuracy" and "loose" to be "less accuracy." At this time, no, I don't have any conjectures about how to use even more precision/accuracy and remain within uniform TC0.
>
> On the other hand, if by "tight" you mean "less accuracy" and by "loose" you mean "more accuracy," then yes, Theorem 14 implies that transformers with less accuracy are also in uniform TC0, and this is not too surprising given what has already shown by others (Table 1).
>
> Does this answer your question? If not, would you please clarify, and I'll do my best to answer?

---

> > ### Comment · Reviewer_gNug · 2024-11-19
> > **Clarification About My Question**
> >
> > Sorry for the confusion. By "loose," I’m referring to the assumption about the absolute error, which seems too challenging to achieve in practice. My original question is whether we can relax this assumption—specifically, use "less accurate" assumptions on the *absolute error*—while still remaining within DLOGTIME-uniform $TC^0$?

---

> > > ### Author Response · Authors · 2024-11-19
> > > **Re: Is $2^{-O(poly(n))}$ too loose?**
> > >
> > > Thanks for the clarification. Yes, Theorem 14 doesn't claim that *every* $\hat{T}$ that approximates $T$ with error at most $\epsilon$ is in $TC^0$; it only asserts there exists such a $\hat{T}$. So if we increase $\epsilon$ to something bigger, the same $\hat{T}$ still has error at most $\epsilon$ and is still in uniform $TC^0$.

---

### Review · Reviewer_g37N · 2024-11-22

**Summary Of Contributions:**

This paper is another addition to establishing the computational complexity and as a consequence the decision problems solved by transformers both of the hard attention (AHAT) and soft attention variety (SMAT)

- The authors aims to improve on previous results i.e AHAT (with limited precision) can be approximated in L-Uniform $TC^0$. The improvement being it is now in DLOGTIME Uniform $TC^0$ with arbitrary precision
- The authors aim to push the previously established complexity class for for SMATS from input floats of **log size** bits  to input floats of **poly size** bits
- The authors are attempting to break ground with expressing the expressiveness of SMAT by abstracting away precision and rounding details and soley focusing on SMAT output approximation error (establishing the absolute error to  be at most $2^{-\textrm{O\(poly\(n)\)}}$

**Audience:**

Yes

**Claims And Evidence:**

Yes

**Requested Changes:**

- Section 5's ending i.e **Corollary 16** definitely needs an expansion
- Under **Definition 9** the definition of a p-bit number perhaps needs more explanation of how the p bits are distributed between the significand and exponent

**Strengths And Weaknesses:**

### Strengths
- For the most part, it is a piece of art
- The paper no doubt is full of profound advances in pushing the limits of what was previously known about the expressiveness of hard and soft attention transformers and attempts to break new ground by using approximation error as opposed to internal details of computation

### Weaknesses
- Proofs end a little too abruptly. There is not enough background knowledge given to connect the dots. Better final statements is very much needed
- It is very easy to lose track of how the technical arguments end up contributing to the lemma/theorem. Cases in point: **Lemma 11** and **Lemma 12** , for all the technical sophistication it seems to **implicitly** boil down to whatever circuit you're looking for as long as the algorithm can be boiled down to these already defined as $TC^0$ circuits and put them together it should be a $TC^0$ circuit
- The implicit reigning concept that **DLOGTIME uniform $TC^0$** circuits are closed under an input-to-output class size preserving, circuit size maintaining composition _is understated_

### Concerns
1. In **Lemma 12**

   > there is an N $ \in O(p)$ that makes the relative error at most $2^{−p−1}$

   I would reason that there needs to be a mention of how the maximum relative error w.r.t precision to give the reader as to how the number was arrived at

 2. **Corollary 16** definitely needs more development.  Maybe it's because of space constraints or something else but atleast to me there's no doubting that Section 5 ends rather abruptly without connecting the dots properly. The claimed easy proof for __Corollary 16__ according to me is very much needed to establish how  $\tilde{T}$ is in $TC_0$

3. Under **Lemma 1** Before _Step 1_ shouldn't the numbers' significands be normalized given the relation $| e_i - e_j| \lt 2p + \lceil \log_2 n \rceil $ excludes the  significands and order of magnitudes can be misinterpreted as we're only looking at the exponents?

4. Under **Definition 9** the definition of a p-bit number perhaps needs a bit of explanation of how the p bits are distributed between the significand and exponent. The lack of this makes **lemma 11** hard to follow

 5. Under **Theorem 14**

    > To simplify the error analysis, all of the above operations are performed on O(poly(n))-bit rational numbers. All operations are exact except √x and exp x (inside softmax), which, by Lemma 12, can be approximated with relative error $\eta$ for any $\eta \in 2^{-\textrm{O\(poly\(n)\)}}$.

    Specifically for $\sqrt{x}$, lemma 12 seems to pertain to floating point numbers which I'm aware is a superset of rational numbers. But are we able to use the same argument here ?

6. Under **Section 5**  how much of the following goal:

    > In this section, we propose an alternative approach, which is to limit the error of the final result of a transformer approximation and abstract away from details (like precision and rounding) of how that level of error is achieved

   is being achieved if in the process of simplifying the error analysis an assumption is being made about the precision?


### Nitpicks and Questions
1. Under **Background** given how important the concept of _uniformity_ is, it is not defined which is rather odd given every other concept like _DLOGTIME_, _transformers_ are defined
2. Under **Section 2.2 Complexity classes**

   > The class $TC_0$ is also the class of  ... (bit about FOL)

   is  perhaps best kept as Remark as it does (please correct me if I'm wrong) nothing for the development of the ideas in this paper (as far as I can tell)
3. Under **Lemma 11**

   > We compute the partitioning into blocks as follows ....

   I would reason that the relation and it's associated transitive closure is already mentioned, this block is perhaps redundant. Perhaps I am missing something?
 4. The complexity class of rounding isn't mentioned anywhere (perhaps I missed this)
 5.  Under **Section 5** in this particular part

      >  and abstract away from details (like precision and rounding) of how that level of error is achieved

      other than *precision* and *rounding*, is there anything else that is being abstracted away?

---

> ### Author Response · Authors · 2024-11-22
> **Connecting the dots**
>
> Thank you very much for your very careful review!
>
> **Weakness 1-2**: To reduce abruptness and keep the big picture clearer, the proofs of Theorem 7 and 13 can certainly be expanded to explain how all the pieces fit together. Theorem 14 could be similarly expanded, and (if the reviewer thinks it would help) it could be broken apart into lemmas for the various operations and a theorem that connects them all together.
>
> **Concern 2 / Change 1**: For Corollary 16, I can add the following proof: Let $L$ be a language, and let $T$ be a SMAT that recognizes $L$ with margin $\epsilon \in 2^{-O(poly(n))}$. By Theorem 14, there is a function $\hat{T}$ in uniform TC0 such that for all $w$, $|\hat{T}(w)-T(w)|\le\epsilon$. For any string $w \in L$, we have $T(w) > \epsilon$, so $\hat{T}(w) \ge \hat{T}(w) - |\hat{t}(w)-T(w)| > \epsilon - \epsilon = 0$. And for any string $w \not\in L$, we have $T(w) < -\epsilon$, so $\hat{T}(w) < 0$. Therefore, $\hat{T}$ also recognizes $L$.
>
> (Response to be continued in separate posts)

---

> > ### Comment · Reviewer_g37N · 2024-11-27
> > **Re:connecting the dots**
> >
> > Apologies for the delay and you're welcome
> >
> > **re:weakness 1-2**: All additions stated will be appreciated for the sake of clarity. That said, the last clause
> > > and (if the reviewer thinks it would help) it could be broken apart into lemmas for the various operations and a theorem that connects them all together.
> >
> > while appreciated (I think)it  is not necessary so I believe we can do without
> >
> > **re:Concern 2/Change 1**: The proposed addition sounds good.
> >
> > (on the side I'm assuming $\hat{t}$ is a typo and I didn't quite get how the $\epsilon - \epsilon$ part is arrived at so I assumed it's my current lack of symbolic dexterity and what you have is a variant of this:
> >
> > 1. $-\epsilon \le \hat{T}(w) - T(w) \le \epsilon $
> > 2. $-\epsilon + T(w) \le \hat{T}(w) - T(w) + T(w) \le \epsilon + T(w) $
> > 3. given $T(w) \gt \epsilon , \epsilon \gt 0$ implies $T(w) - \epsilon \gt \epsilon - \epsilon $ implies $T(w) - \epsilon \gt 0$
> > 4. $ 0 \lt \hat{T}(w) \le \ldots $

---

> ### Author Response · Authors · 2024-11-22
> **DLOGTIME-uniform TC0, composition closure, and FOM[BIT]**
>
> **Question 1**: The definition of DLOGTIME-uniform TC0 is intentionally brief, since it's defined in many other places, but I can expand on the concept of uniformity. In particular, I agree that the mention of "the circuit for length n" is mysterious and needs explanation.
>
> **Weakness 3**: That DLOGTIME-uniform TC0 functions are closed under composition is asserted by [1], and is easier to see, I think, if we view them as formulas of FOM[BIT].
>
> A function f from numbers with log(n) bits to numbers with log(n) bits is defined by a formula F(x,y), which is true just in case y = f(x). If f and g are defined by F(x,y) and G(y,z), then their composition gf is defined by the formula $GF(x,z) = \exists y [F(x,y) \land G(y,z)]$.
>
> An n-bit number a is defined by a formula A(i), which is true just in case the i-th bit of a is 1. We could think of a function f from n-bit numbers to n-bit numbers as being defined by a "template formula" F[X], written in terms of a predicate X(i) that defines an n-bit number. If b = f(a), then b is defined by the formula B(i) obtained by taking F[X] and replacing occurrences of X(i) with A(i). If c = g(b), where g is similarly defined by a "template formula" G[Y], then c is defined by the formula C(i) obtained by taking G[Y] and replacing occurrences of Y(i) with B(i).
>
> **Question 2**: If the above argument is included in the paper, then hopefully it's clear why FOM[BIT] deserves to be mentioned.
>
> [1] Emil Jerabek. Root finding with threshold circuits. https://arxiv.org/pdf/1112.3925
>
> (Response to be continued)

---

> ### Author Response · Authors · 2024-11-22
> **Response to other comments on Section 4**
>
> Concern 1: In Lemma 12, you asked where $N$ comes from. Because the absolute error is in $O(|r-1|^N)$, there is some $c$ such that the relative error is at most $c|r-1|^N/\sqrt{r} \le 2c(3/4)^N$. Then set $2c(3/4)^N = 2^{-p-1}$ and solve for $N$ to get $N = (p+2+\log c) / \log (4/3) \in O(p)$. I hope that helps; does the derivation of the error term $O(|r-1|^N)$ also need to be explained?
>
> Concern 3. In Lemma 11 before Step 1, yes, since the $\langle m_i, e_i\rangle$ are floating-point numbers, by Definition 9 they must be normalized, that is, $|m| \in [2^{p-1},2^p)$.
>
> Concern 4 / Change 2. In Definition 9, the constraints on $m$ and $e$ imply that the significand $m$ has $(2^p-2^{p-1}) \cdot 2+1 = 2^p+1$ possible values and can be stored with $p+1$ bits, and the exponent has $2^p-(-2^p) = 2^{p+1}$ possible values and can be stored with $p+1$ bits. If it's annoying for a "p-bit" floating-point number to actually have $2(p+1)$ bits, I can change the definition to make it have $p/2$ bits for the significand and $p/2$ bits for the exponent.
>
> Question 3. Although the $\sim$ relation is defined at the beginning of Step 1, it isn't obvious (to me) how this relation and its transitive closure would be computed in uniform TC0. (Transitive closure is not known to be in uniform TC0.) So I think the paragraph starting "We compute the partitioning into blocks as follows" is needed, and I'll modify it to read, "We can compute the partitioning into blocks in uniform TC0 as follows."
>
> Question 4. You're right that the paper doesn't show that $round_p(\langle m, e\rangle)$ is in uniform TC0. I can add an explanation that, in uniform TC0, we can compute a shift $q = \lfloor \log |m| \rfloor - p + 1$ (Hesse, 2002) and return the normalized floating-point number $\langle m / 2^q, e + q\rangle$.

---

> > ### Comment · Reviewer_g37N · 2024-11-27
> > **Re:Response to other comments on Section 4**
> >
> > **re:Concern 1**: That works and no the error term doesn't need to be derived but I appreciate the ask
> >
> > **re: Concern 3**: My apologies the definition is pretty clear. This was an oversight on my part
> >
> > **re: Concern 4 / Change 2** I see, just a note (if that makes sense) would suffice. I would reason this is perfect place for the O notation i.e  O(p) bits (and the user deduces the details of the division as you have) but you let me know as I could be completely incorrect about the same
> >
> > **re:Question 3**: Point taken and I believe your addition adds clarity
> >
> > **re:Question 4**: Thank you for the clarification and again I think the addition will be appreciated by any reader

---

> ### Author Response · Authors · 2024-11-22
> **Response to other comments on Section 5**
>
> Concern 5. Yes, I agree that since the Taylor approximation of $\sqrt{x}$ makes use of the floating-point representation of $x$, it's not obvious how this would work on rational numbers. I can add an explanation that in uniform TC0, we can compute $k = \lfloor \log x\rfloor + 1$ if $\lfloor \log x\rfloor$ is odd, $k = \lfloor \log x \rfloor+2$ otherwise (Hesse, 2002), let $r = x/2^k$, and proceed as in Lemma 12.
>
> Concern 6. If I understand the question correctly, the statement of Theorem 14 does not make mention of any numeric representation, and the use of $O(poly(n))$-bit rational numbers is not an additional assumption. Rather, the statement of Theorem 14 imposes a requirement on $\hat{T}$, and in the proof we are free to implement $\hat{T}$ in any way as long as it meets this requirement. The choice of $O(poly(n))$-bit rational numbers, then, is just an implementation decision.
>
> Question 5. Yes, Theorem 14 also doesn't specify whether numbers are represented as floating-point, fixed-point, rational, etc. Although, Theorem 14 mentions a real-valued function being computable in TC0, without specifying what that means. If that is a problem, I can change it.

---

> > ### Comment · Reviewer_g37N · 2024-11-27
> > **re:Response to other comments on Section 5**
> >
> > **re: Concern 5** : Understood and please do add that in
> >
> > **re: Concern 6**: I see. I stand corrected
> >
> > **re: Question 5**: Understood and no it is not a problem

---

> ### Comment · Reviewer_g37N · 2024-11-27
> **re: DLOGTIME-uniform TC0, composition closure, and FOM[BIT]**
>
> **re:Question 1**: I understand what you mean
>
> **re:Weakness 3** AND **re:Question 2**:  Thank you for the explanation and the linked paper. I think the addition will do the trick. On a related note, I wonder if just referring readers to this line
> > TC0 functions are closed under composition, and under “parallel execution”
>
> in the paper by Emil Jerabek can do away with any mention of FOM[BIT]  (including the proposed addition above).
> Just to be clear either should work as I think you all know best
>
> Just for my curiosity, does an explicit exclusion of the _parallel execution_ clause change the validity of composition of $TC^0$ functions?

---

### Review · Reviewer_dNpr · 2024-11-25

**Summary Of Contributions:**

This is a theoretical paper studying the transformer and the relationship with $TC^0$. It is shown that with suitable approximation and approximation structure, a DLOGTIME-uniform approximation can be achieved. However, the significance of DLOGTIME-uniform $TC^0$ in this context remains unclear. Nevertheless, the paper is well-written and easy to follow.

**Audience:**

Yes

**Broader Impact Concerns:**

The results are primarily theoretical, and their broader impact remains unclear. It is also uncertain whether these theorems represent their optimal form or if there is potential for future refinement and improvement.

**Claims And Evidence:**

No

**Requested Changes:**

1. Provide the argument that meaning of different approximation approaches. (O(log n) and O(poly(n)) )
2. It is essential to examine whether the theorems presented in Table 1 can be extended to multi-layer architectures. This extension would require careful consideration of the interactions and dependencies across layers, as well as potential changes in the underlying assumptions or structures. If such a generalization is feasible, it would significantly enhance the scope and applicability of the results. However, the current formulation may not provide sufficient details or tools to directly support this extension, leaving room for further exploration in future work.

**Strengths And Weaknesses:**

Strengths:
The strength of this work lies in the clear presentation of the results, especially through the use of a table (Table 1). Most of the statements are well-articulated and easy to understand.

Weaknesses:
1. It is hard to identify the studying scenario of approximation of different precision such as $O(\log n)$ and $O(\textrm{poly}(n))$. Usually, when a model is trained, it is unlikely people will change the precision of weight or computation based on the input sequence length. It is necessary that the author give interpretation of this modification in the paper. (As can be seen, all other paper are discussing $O(\log n)$ scenario)
2. There is no discussion on the challenges of extending the results to multi-layer cases. It is common for results in multi-layer settings to differ from those in single-layer scenarios. Including a brief discussion or sketch of a potential approach to study the multi-layer case would be beneficial for readers and could guide future research.

---

> ### Author Response · Authors · 2024-11-25
> **Precision**
>
> Thank you for your review. I'm afraid I don't understand what you mean by "the studying scenario." If you are asking about what practical situations people might use O(log n) or O(poly(n)) precision in, I would give two answers.
>
> First, I actually do think that making precision depend on sequence length is a reasonable thing to do. Real-world uses of language models use many different levels of precision, starting with 32 or 16 bits and often quantized down to 8 bits or even all the way down to 1 bit. It's not hard to imagine that input sequence length would be one of the criteria used for choosing what level of precision to use.
>
> Second, as discussed in the Conclusion and Limitations section, this is a purely theoretical paper. I admit it's hard to imagine a scenario where one would be willing to use O(n) bits of precision, and it's much harder to imagine $O(n^k)$ bits for k>1. But the goal is not to implement these ultra-high precision transformers; the goal is to study theoretical limitations of transformers. It might be thought that previous results depend in a crucial way on the assumption of O(log n) bits, and this paper demonstrates that they do not; even with absurdly high precision, transformers are still limited to uniform TC0.
>
> I hope that helps. If I have not addressed your concerns about precision, would you please clarify your concerns, and I will try again?

---

> ### Author Response · Authors · 2024-11-25
> **Depth**
>
> All of the results in Table 1, both the previously published ones and the ones in the present paper, are for transformers with an arbitrary number of layers.

---

> ### Author Response · Authors · 2024-11-25
> **Claims and Evidence**
>
> In answer to the question, "Are the claims made in the submission supported by accurate, convincing and clear evidence?" you answered no. If you could clarify which claims made in the paper are not supported, I would be happy to respond.

---

> > ### Comment · Reviewer_dNpr · 2024-12-16
> >
> > Thank you for your detailed and thoughtful responses to my comments. After considering your explanations, I now better understand the theoretical significance of your work and its broader implications for understanding transformer models.

---

### Author Response · Authors · 2024-12-03
**Version 2 uploaded**

I've uploaded a new version that I believe addresses all the reviewers' suggestions. Please let me know if there are further changes that are required. Thank you!

---

> ### Comment · Reviewer_g37N · 2024-12-05
> **Re:version 2**
>
> Thank you for the additions and looks splendid to me

---

### Decision · Action_Editor_TKSD · 2024-12-31

**Recommendation:** Accept as is

**Comment:**

This paper builds on previous results that aim to characterize Transformers in terms of their membership in complexity classes when taking into consideration the model's precision. The paper unifies these past findings and extends them a bit, in particular showing DLOGTIME-uniform TC0 complexity for a few different Transformer variaitons (including different versions of attentions and precision). All reviewers felt the results were sound and possibly of interest and therefore recommended acceptance.

**Audience:**

The paper has nontrivial results that may be of interest to TMLR audience members that are concerned with theoretical/complexity limitations and characterizations of Transformers.

**Claims And Evidence:**

This paper is primarily theoretic; reviewers did not raise any (post-rebuttal) concerns as to the validity of the proofs or theoretical claims.